

# MetaCRAST: reference-guided extraction of CRISPR spacers from unassembled metagenomes

Abraham G. Moller and Chun Liang

Department of Biology, Miami University, Oxford, OH, United States of America

## ABSTRACT

Clustered regularly interspaced short palindromic repeat (CRISPR) systems are the adaptive immune systems of bacteria and archaea against viral infection. While CRISPRs have been exploited as a tool for genetic engineering, their spacer sequences can also provide valuable insights into microbial ecology by linking environmental viruses to their microbial hosts. Despite this importance, metagenomic CRISPR detection remains a major challenge. Here we present a reference-guided CRISPR spacer detection tool (**Meta**genomic **C**RISPR **R**eference-**A**ided **S**earch **T**ool—MetaCRAST) that constrains searches based on user-specified direct repeats (DRs). These DRs could be expected from assembly or taxonomic profiles of metagenomes. We compared the performance of MetaCRAST to those of two existing metagenomic CRISPR detection tools—Crass and MinCED—using both real and simulated acid mine drainage (AMD) and enhanced biological phosphorus removal (EBPR) metagenomes. Our evaluation shows MetaCRAST improves CRISPR spacer detection in real metagenomes compared to the *de novo* CRISPR detection methods Crass and MinCED. Evaluation on simulated metagenomes show it performs better than *de novo* tools for Illumina metagenomes and comparably for 454 metagenomes. It also has comparable performance dependence on read length and community composition, run time, and accuracy to these tools. MetaCRAST is implemented in Perl, parallelizable through the Many Core Engine (MCE), and takes metagenomic sequence reads and direct repeat queries (FASTA or FASTQ) as input. It is freely available for download at https://github.com/molleraj/MetaCRAST.

# INTRODUCTION

The clustered regularly interspaced short palindromic repeat (CRISPR) arrays found in prokaryotic genomes can help us better understand viral-microbial interactions important in many ecosystems. Viruses can release cellular nutrients back into the ecosystem through lytic infection, forming an ecological short-circuit called the viral shunt (*Weitz & Wilhelm, 2012*). In this manner, viruses not only contribute to nutrient cycling in individual ecosystems, but also to maintaining biogeochemical cycles on a broader scale. The short spacers of viral DNA incorporated into CRISPR arrays form a historical record of past infections, thus linking virus to host (*Sorek, Kunin & Hugenholtz, 2008*; *Makarova, Wolf*

Corresponding author
Chun Liang, liangc@miamioh.edu

& Koonin, 2013). This power of CRISPR spacers to determine viruses' host specificity has recently been exploited using metagenomes from many ecosystems (*Anderson, Brazelton & Baross, 2011*; *Sanguino et al., 2015*; *Edwards et al., 2015*). While many tools exist for detecting CRISPRs in assembled genomes (*Bland et al., 2007*; *Edgar, 2007*; *Grissa, Vergnaud & Pourcel, 2007a*; *Rousseau et al., 2009*), few exist for CRISPR detection in metagenomic reads (*Rho et al., 2012*; *Skennerton, Imelfort & Tyson, 2013*; *Skennerton, 2006*).

The repetitive nature of CRISPRs makes them difficult to assemble from metagenomes, necessitating special tools to detect them in unassembled reads. Several tools have been developed to detect and assemble CRISPR arrays in unassembled reads rather than assembled contigs. The tool MinCED (Mining CRISPRs in Environmental Datasets), like metaCRT (*Rho et al., 2012*), is a modified version of CRT (*Bland et al., 2007*) that detects CRISPR spacers (*Skennerton, 2006*), while the tool Crass (CRISPR assembler) detects and assembles CRISPR arrays (*Skennerton, Imelfort & Tyson, 2013*), both from raw metagenomic reads. MinCED searches each read for CRISPRs using the same strategy as CRT; it searches for appropriately spaced short k-mers from which it extends longer repeats if appropriately frequent nucleotides are identified at the ends of the growing repeats. Crass relies on a hybrid algorithm to detect spacers that blends strategies of CRT (*Bland et al., 2007*) and CRISPRFinder (*Grissa, Vergnaud & Pourcel, 2007b*). In long reads (>177 bp), it searches for repeats using the CRT strategy previously described. In short reads (<177 bp), on the other hand, it searches for appropriately spaced full-length repeats (i.e., 20–50 bp) and extends these repeats only with identical nucleotides, thus avoiding the potential errors caused by the CRT algorithm over- or under-extending the few repeats found in a short sequence. Crass then searches further for reads containing a single repeat, determines consensus direct repeats, uses the first and last k-mers of detected spacers to build a graph of spacer arrangement, and assembles CRISPR arrays using this graph. Both MinCED and Crass do not rely on prior knowledge of direct repeat sequences, making them *de novo* detection methods. Instead, they use heuristics to determine whether detected repeats are indeed CRISPRs. Such heuristics include threshold array lengths to avoid short, spurious CRISPR arrays and threshold repeat-spacer similarities to avoid arrays where spacers are too similar to repeats (*Bland et al., 2007*; *Grissa, Vergnaud & Pourcel, 2007a*; *Skennerton, Imelfort & Tyson, 2013*), which might indicate microsatellites rather than CRISPRs.

In this work, we present the Metagenomic CRISPR Reference-Aided Search Tool (MetaCRAST), a novel reference-guided tool to improve CRISPR spacer detection in unassembled metagenomic sequence reads. While MetaCRAST, to our knowledge, is the first reference-guided, read-dependent metagenomic CRISPR detection tool, prior studies have used known direct repeats to improve CRISPR detection. The genomic CRISPR identification algorithm CRISPRDetect matches newly identified direct repeats to a reference library to refine repeat boundaries and validate arrays (*Biswas et al., 2016*). Searching reference repeat libraries, together with annotating *cas* genes adjacent to CRISPR arrays, has been used to exclude false positive "putative" CRISPRs from CRISPR annotation (*Zhang & Ye, 2017*). Unlike MinCED and Crass, as a reference-guided method, MetaCRAST constrains spacer detection by searching metagenomes for direct repeats (DRs) that the user specifies. Relationships amongst these tools and such differences in
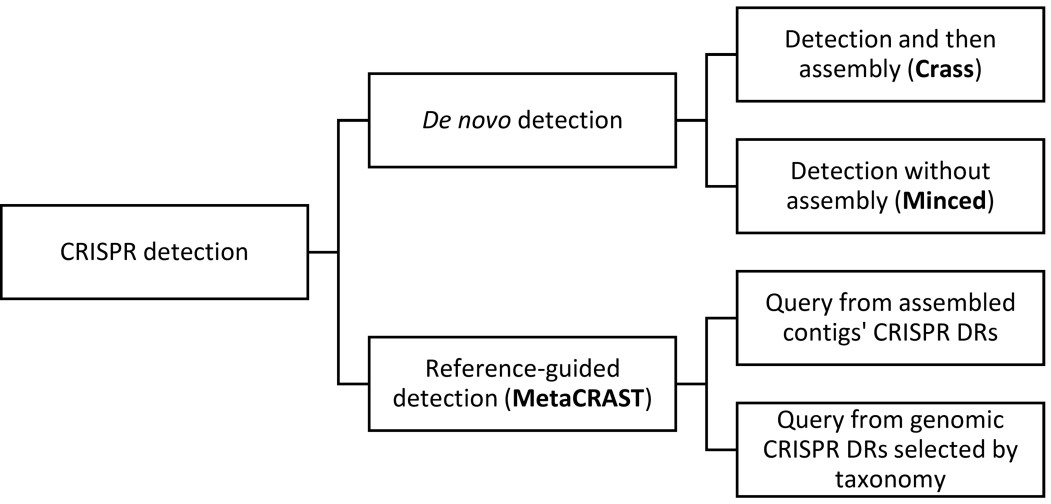

**Figure 1** **This diagram outlines relationships amongst different metagenomic CRISPR detection methods.** CRISPR detection can be performed either using specified direct repeats (reference-guided detection) or without prior knowledge of direct repeat sequences (*de novo* detection). *De novo* detection searches raw metagenomic reads for direct repeat sequences of the appropriate length and spacing (i.e., 25–60 bp long repeats with 25–60 bp spacers between them). *De novo* detection techniques either detect spacers in reads only (MinCED) or assemble reads into arrays (Crass). Reference-guided CRISPR detection, on the other hand, searches reads for user-specified direct repeat sequences, and extracts spacers from between direct repeat sequences identified in reads containing direct repeats. While the query is user-specified, general strategies for generating a query include using direct repeats found in assembled metagenomic contigs with CRISPR array detection tools (e.g., PILER-CR) or direct repeats found in genomic CRISPR arrays (e.g., those found in microbial genomes included in CRISPRdb) that might be expected based on taxonomic profiles. An example of the latter strategy would be searching for known genomic *Streptococcus pyogenes* direct repeats if *Streptococcus pyogenes* is found in the metagenome's taxonomic profile.

use are further illustrated in Fig. 1. Such specified DRs may be selected based on assembly or taxonomic profiling of metagenomic reads. MetaCRAST improves CRISPR annotation by allowing users to control for the taxonomic composition of the metagenome. It also avoids the rejection of true CRISPRs that can occur due to the heuristics required for *de novo* detection methods. In addition, unlike Crass and MinCED, MetaCRAST provides consistent performance over different read length Illumina datasets.

## MATERIALS AND METHODS

### Algorithm and implementation

MetaCRAST can constrain spacer detection by expected host species' DRs or DRs identified from assembly (Fig. 2A). It searches each read for DR sequences matching query DRs specified by the user. These DRs can be selected from CRISPR arrays detected with genomic CRISPR detection tools such as PILER-CR (*Edgar, 2007*), CRF (*Wang & Liang, 2017*), or CRISPRFinder (*Grissa, Vergnaud & Pourcel, 2007b*) in fully assembled microbial genomes or assembled metagenomic contigs. The steps of the MetaCRAST pipeline are outlined in Fig. 2B. In the first step of the pipeline, reads containing DRs within a certain Levenshtein edit distance (i.e., number of insertions, deletions, or substitutions necessary

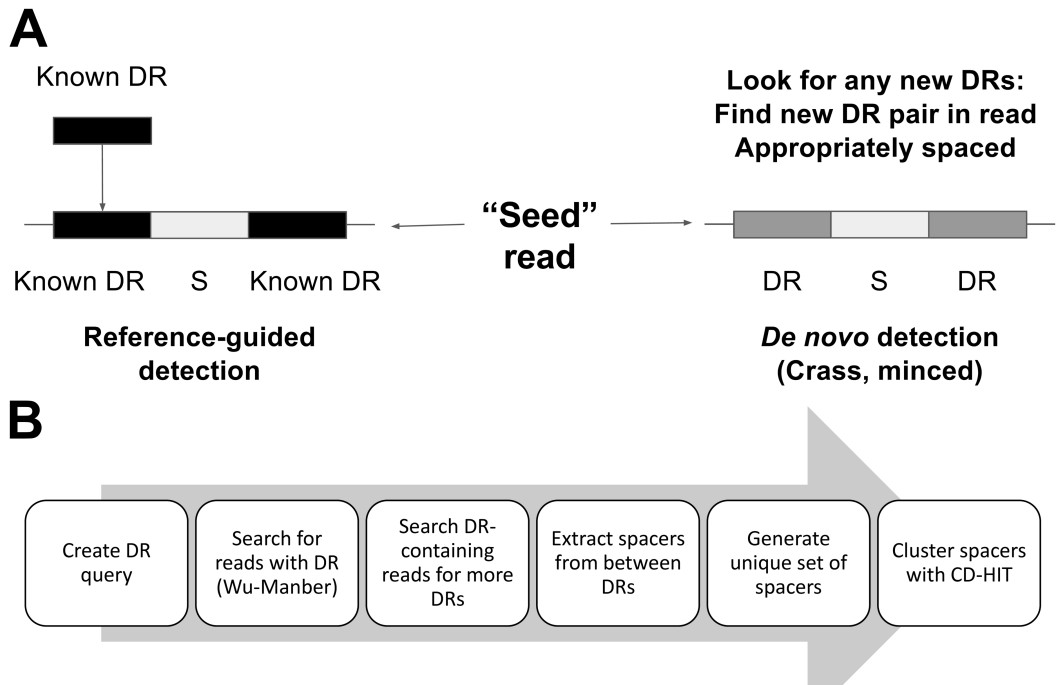

**Figure 2   A comparison of per-read CRISPR detection strategies (A) between MetaCRAST and existing *de novo* detection tools (e.g., Crass, MinCED) and an outline of the MetaCRAST workflow (B).** DR represents direct repeat, while S represents spacer.

to convert one sequence to another) of the query DRs are quickly identified using the Wu-Manber multi-pattern search algorithm (*Wu, Manber & Myers, 1995*). In the second step, individual reads found to contain a query DR sequence are searched for two or more copies of the query DRs. In the third step, the sequence fragments between the DRs detected in these sequence reads are extracted into a comprehensive spacer set, which are then clustered using CD-HIT into a non-redundant unique spacer set stored in FASTA format (*Li & Godzik, 2006*).

MetaCRAST is implemented in Perl as a command line tool to analyze metagenomes in FASTA or FASTQ formats. The tool has been implemented in several versions that differ in metagenome loading method (using BioPerl or readfq, the latter of which was paired either with the standard open routine to load a single file or mce_open for parallel file loading). Optionally, the user can specify the maximum spacer length, the distance metric used for comparing DRs to reads (Hamming or Levenshtein), whether to search for the reverse complement of the DR, the CD-HIT similarity threshold for clustering spacers, and the maximum number of threads to use to parallelize the search. The reverse complement argument (-r) should be used when the CRISPR direction is unknown. When the search is run in parallel, the FASTA (or FASTQ) file is split based on the specified number of threads. All command line arguments are further described in Table 1. Each split file is searched in parallel. An additional tool has been provided to assist taxonomy-guided query

**Table 1  Command line arguments for MetaCRAST.** Required arguments are in bold.

| Argument | Description |
|---|---|
| **-p** | Pattern file containing query DR sequences in **FASTA or FASTQ** format |
| **-i** | Input metagenome in **FASTA or FASTQ** format |
| **-o** | Output directory for detected reads and spacers |
| **-d** | Allowed edit distance (insertions, deletions, or substitutions) for initial read detection with the Wu-Manber algorithm and subsequent DR detection steps |
| -t | Temporary directory to put metagenome parts (use this if -n option also selected) |
| -q | Input metagenome is a FASTQ file (directs use of fastq-splitter.pl instead of fasta-splitter.pl) |
| -h | Use Hamming distance metric (substitutions only - no insertions or deletions) to find direct repeat locations in reads (default: use Levenshtein distance metric - look for sequences matching DR within insertion, deletion, and/or substitution edit distance) |
| -r | Search for reverse complement of CRISPR direct repeat sequences |
| -l | Maximum spacer length in bp |
| -c | CD-HIT similarity threshold for clustering spacers detected for each query direct repeat (value from 0 to 1) |
| -a | CD-HIT similarity threshold for clustering all detected spacers (value from 0 to 1) |
| -n | Number of processors to use for parallel processing (and number of temporary metagenome parts) |

selection. This tool searches a taxonomically-annotated library of CRISPRdb DRs for those that belong to a particular taxon query (e.g., *Streptococcus*).

To analyze the distribution of taxonomic affiliations to direct repeats, we examined all direct repeats found in microbial genomes using the CRISPRdb database. CRISPRdb provides a library of direct repeats labeled with respective GenBank accessions in the CRISPR utilities section of the database (*Grissa, Vergnaud & Pourcel, 2007a*). We processed this library to assign taxonomy information based on GenBank accession. Taxonomy information was extracted from GenBank records with the Perl module Bio::DB::GenBank. Statistics describing the distribution of unique binomial names or genuses to which individual direct repeats affiliated was compiled with Microsoft Excel. Binomial name (species-level) and genus statistics are presented in Table 2.

## Performance evaluation with simulated and real metagenomes

To study the relationship between CRISPR spacer detection and read length or sequencing technology, simulated acid mine drainage (AMD) and enhanced biological phosphorus removal (EBPR) metagenomes were generated using Grinder (*Angly et al., 2012*). We generated simulated metagenomes over a range of average read lengths (100 to 600 base pairs) using models of 454 (*Balzer et al., 2010*) and Illumina (*Korbel et al., 2009*) errors. Following previous studies, we used a fourth-degree polynomial ($3e-3 + 3.3e-8 * i^4$, where i is the nucleotide position from the 5′ end of the read, and the output is percentage chance of an error at that position) to model the Illumina sequencing error rate (*Dohm et al., 2008*; *Korbel et al., 2009*; *Angly et al., 2012*). This polynomial determined the probability of substitution, insertion, or deletion at each base of a simulated read (*Korbel et al., 2009*). For Illumina simulations, the ratio of substitutions to insertions and deletions was set to

**Table 2  Distribution statistics for binomial name and genus-level taxonomic affiliation of CRISPRdb direct repeats.** A library of direct repeats labeled with respective GenBank accessions from CRISPRdb was processed to assign taxonomy information based on GenBank accession. Taxonomy information was extracted from GenBank records with the Perl module Bio::DB::GenBank. Statistics describing the distribution of binomial names or genuses to which individual direct repeats affiliated were compiled with Microsoft Excel.

| Statistic | Binomial names | Genuses |
|---|---|---|
| Mean | 1.308 | 1.063 |
| Median | 1 | 1 |
| Mode | 1 | 1 |
| Minimum | 1 | 1 |
| Maximum | 46 | 20 |
| Standard deviation | 1.567 | 0.521 |

80:20 by default. For 454 metagenome simulations, we modeled homopolymer errors as homopolymer length variation within simulated reads. The distributions of homopolymer lengths were defined by the mean n and standard deviation $0.03494 + n * 0.06856$, where n is the homopolymer length, based on a prior study (*Balzer et al., 2010*; *Angly et al., 2012*).

We generated six simulated metagenomes per condition (average read length, model, and microbial community). We used highly simplified taxonomic profiles to model the AMD and EBPR metagenomes (Tables S1 and S2). To test the effects of community composition on spacer detection, we simulated the AMD metagenome with a 454 error model and 600 bp average read length, varying the relative proportions of *Leptospirillum* and *Ferroplasma* genome used for the simulation (i.e., from 0 to 100% *Leptospirillum*). All simulated metagenomes contained 100,000 reads. 454 metagenomes were generated with this command: *grinder -reference_file AMDgenomes.fasta—abundance_file AMDprofile.txt -total_reads 100000 -read_dist (one of 100, 150, 200, 250, 300, 400, or 600) normal 50 -homopolymer_dist balzer*. All 454 read length distributions were normal with a standard deviation of 50 bp. Illumina metagenomes were generated with this command: *grinder -reference_file AMDgenomes.fasta -abundance_file AMDprofile.txt -total_reads 100000 - read_dist (one of 100, 150, 200, 250, or 300) -md poly4 3e−3 3.3e−8*. All Illumina read length distributions were uniform with all reads having exactly the average read length.

Simulated metagenomes were searched for CRISPR spacers using Crass (*Skennerton, Imelfort & Tyson, 2013*), MinCED (Skennerton), and MetaCRAST. Crass and MinCED were run with default parameters (*crass grinder-reads.fa; minced -spacers grinder-reads.fa minced.crispr*). The default minimum and maximum DR lengths for both Crass and MinCED were 23 and 47 bp. The default minimum and maximum spacer lengths for both Crass and MinCED were 26 and 50 bp. MetaCRAST was run with a taxonomy-guided query (Tables S3 and S4), a maximum spacer length of 60, a maximum allowed edit distance (insertions, deletions, or substitutions) between query and target direct repeats of three, a CD-HIT clustering similarity threshold of 0.9, and a total of 16 parallel threads (*MetaCRAST -p query.fa -i grinder-reads.fa -o MetaCRAST -d 3 -l 60 -c 0.90 -a 0.90 -n 16 -t tmp*). We selected a maximum allowed edit distance of three based on results of our prior metagenomic CRISPR detection studies, which showed MetaCRAST searches with a
taxonomy-guided query that found similar numbers of spacers to Crass when we set this edit distance (*Moller & Liang, 2017*). For all analyses, detected spacers were clustered with CD-HIT with a similarity threshold of 0.9 (*cdhit -i spacers.fa -o spacersCD90.fa -c 0.9*) to reduce spacer redundancy. Performance on these simulated metagenomes was evaluated based on total number of spacers detected, number of false positive spacers detected, and run time for each average read length. For the mixed composition simulated AMD metagenomes described above, spacers were aligned against CRISPR spacers present in the source *Leptospirillum* and *Ferroplasma* genomes and the number of matching true positive spacers for each organism were reported.

The number of false positive spacers found in simulated metagenomes was determined by comparing the total detected spacers with the expected CRISPRdb spacers found in the source genomes used for the simulations (AMD and EBPR). Alignments were made to the annotated CRISPRdb spacers using BLAST with an *E*-value cutoff of 1e−6 (*Altschul et al., 1990*). This analysis was repeated with an *E*-value cutoff of 1e−1 to consider whether the original threshold was too stringent. The number of detected spacers that were aligned to expected ones was subtracted from the total number of spacers detected to determine the number of false positive spacers for a particular method and condition. Cases where zero spacers were detected in a metagenome were treated as zero false positive spacers and included in overall analysis. Run times were determined for each metagenome and method using the built-in Linux command *time*. Run time was calculated as the sum of the user and system time (together the total CPU time).

Similarly, CRISPR spacers were also detected by the aforementioned three tools in real AMD and EBPR metagenomes (Table S5) downloaded from iMicrobe (*Hurwitz, 2014*) and taxonomically profiled with MetaPhyler (*Liu et al., 2011*). MetaCRAST analyses of the real metagenomes were performed with taxonomy- or assembly-guided query DRs generated as follows. To make an assembly-guided query, CAP3-assembled contigs (*Huang & Madan, 1999*) were searched for CRISPR DRs using PILER-CR (*Edgar, 2007*), which finds CRISPRs in assembled genomes or contigs. These DRs formed an assembly-guided query (Tables S6 and S7), while DRs found in assembled *Leptospirillum* (AMD), *Ferroplasma* (AMD), and Candidatus *Accumulibacter phosphatis* (EBPR) genomes included in CRISPRdb (*Grissa, Vergnaud & Pourcel, 2007a*) formed a taxonomy-guided query (Tables S3 and S4). All of these aforementioned taxa were found to be major components of the microbial community based on the AMD and EBPR taxonomic profiles determined with MetaPhyler (Tables S8 and S9).

## RESULTS

### Effects of read length, sequencing technology, and community composition on CRISPR spacer detection

We first investigated the relationships between detected spacers and read length or sequencing technology. Performance, here determined by the number of spacers detected, increased with read length over all 454 tests (Fig. 3). While the total number of spacers detected by Crass and MetaCRAST converged as read length increased, the total number

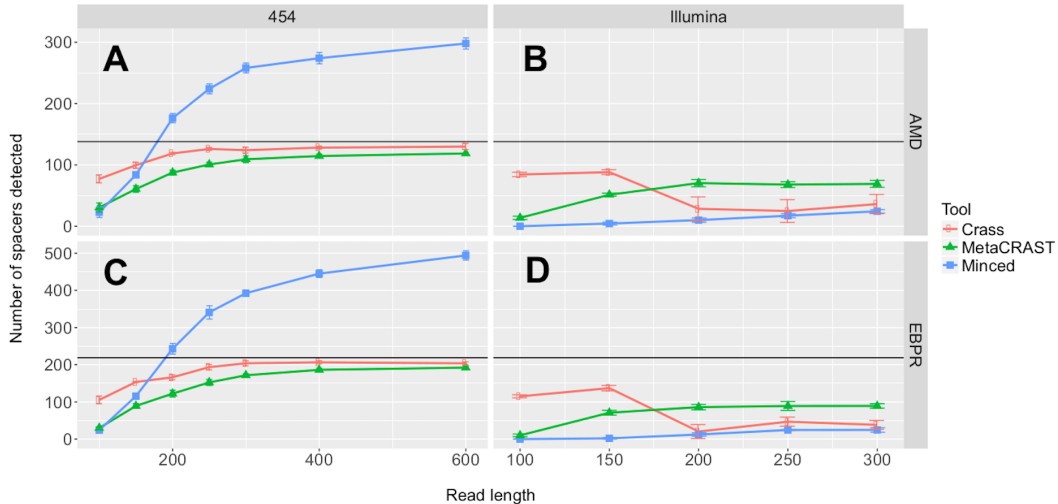

**Figure 3** **Evaluation of MetaCRAST, Crass, and MinCED performance on simulated AMD (A and B) and EBPR (C and D) metagenomes.** The procedure used to generate the simulated metagenomes is described in Materials and Methods. All data points represent the averages of six individual simulations and are presented with error bars representing two times the standard error above and two below the average. The true number of spacers expected in each simulated metagenome is marked with a black line (138 expected in the AMD metagenomes; 219 in the EBPR metagenomes).

of spacers detected by MinCED steadily increased even beyond the true number of spacers found in the genomes used to generate the simulated metagenomes. We speculate that MinCED inconsistently determined DR lengths amongst different CRISPR-containing reads due to its CRT-based algorithm, leading to the same spacers being inappropriately truncated or extended. Meanwhile, amongst metagenomes simulated with the Illumina model, MetaCRAST detected significantly more spacers than Crass and MinCED for average read lengths of 200 bp or greater (Fig. 3; $p < 0.05$ for both AMD and EBPR simulations using unpaired $t$-tests). Crass detected more spacers than MinCED and MetaCRAST for short Illumina reads (100 and 150 bp), however (Fig. 3; $p < 0.05$ for both AMD and EBPR simulations using unpaired $t$-tests).

We also tested the effects of community composition on CRISPR detection for each of the three methods using AMD metagenomes simulated with a 454 error model and 600 bp average read length. We selected the 600 bp average read length for all mixed metagenomes to minimize differences in detection between methods based on read length (Fig. 3). We varied the relative abundances of *Leptospirillum* and *Ferroplasma* from 0 to 100 percent in our taxonomic profiles, thus varying the proportions of CRISPR arrays specific to each included in the simulated metagenomes. For all detection methods, detected spacers specific to a genome decreased as the relative proportion of that taxon decreased, with roughly the same pattern for each method (Fig. 4). As in the read length studies, MinCED consistently detected far more genome-specific spacers in the metagenomes than were originally present in the source genomes (Fig. 4). This may account for its steeper increase in detected genome-specific spacers as the proportion of the corresponding genome in the simulated metagenomes increased.

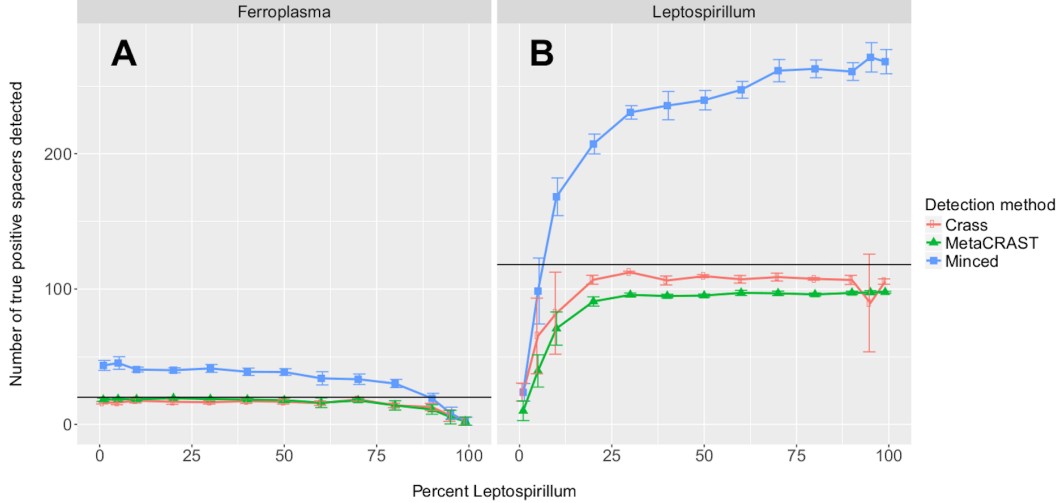

**Figure 4** **Evaluation of MetaCRAST, Crass, and MinCED performance on simulated metagenomes with varying proportions of *Ferroplasma acidarmanus* fer1 and *Leptospirillum* sp. Group II 'CF-1' genome sequences.** Simulated metagenomes were generated with Grinder. The data points shown represent the average number of "true positive" spacers detected that matched spacers in corresponding *Ferroplasma* or *Leptospirillum* CRISPR arrays (A and B, respectively). All data points represent the averages of six individual simulations and are presented with error bars representing two times the standard error above and two below the average. The true number of spacers expected for each genome is marked with a black line (20 expected in the *Ferroplasma* genome; 118 in the *Leptospirillum* genome).

## Evaluation of CRISPR spacer detection on real AMD and EBPR metagenomes

We also evaluated MetaCRAST against Crass and MinCED using real AMD and EBPR metagenomes (*Tyson et al., 2004*; *Martín et al., 2006*). While taxonomy-guided queries consistently found fewer spacers than the other two methods (583 compared to 2,486 for Crass and 4,265 for MinCED in the AMD metagenome; 196 compared to 1,014 for Crass and 1,821 for MinCED in the EBPR metagenome), an assembly-guided MetaCRAST search identified more spacers than Crass did in the AMD metagenome (2,813 compared to 2,486—Fig. 5A). In both AMD and EBPR metagenomes, many common spacers were detected amongst Crass, MetaCRAST (assembly-guided query), and MinCED (7.1% of all detected spacers for AMD and 2.5% for EBPR—Figs. 5B and 5C). Despite this, there were also many spacers detected with Crass and MinCED not identified with MetaCRAST searches (Figs. 5B and 5C). Notably, however, none of the spacers detected with MetaCRAST using the taxonomy-guided query overlapped with the Crass-detected spacers (Figs. 5B and 5C), suggesting MetaCRAST can detect spacers missed by Crass given an appropriate taxonomy-guided query.

## Evaluation of accuracy and runtime performance

In addition to our studies comparing detected spacers over a variety of conditions, we evaluated all three detection methods for spacer detection accuracy and run time (Figs. 6 and 7). We performed these evaluations on the simulated AMD and EBPR metagenomes previously used to examine effects of read length and sequencing technology on CRISPR
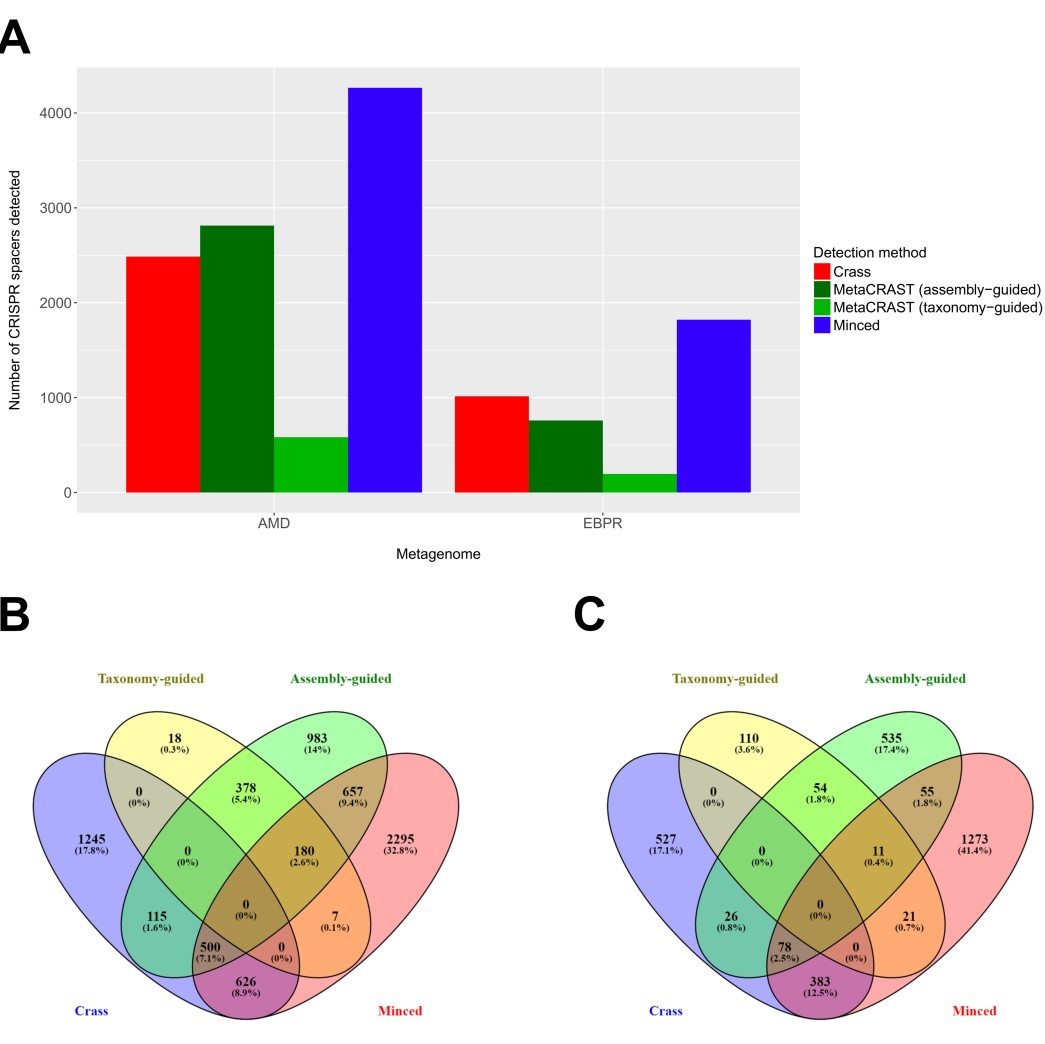

**Figure 5 Evaluation of MetaCRAST, Crass, and MinCED on real AMD and EBPR metagenomes.** (A) Total number of CRISPR spacers detected in real AMD and EBPR metagenomes using four different detection methods—Crass (*de novo*), MetaCRAST (using assembly-guided queries), MetaCRAST (using taxonomy-guided queries), and MinCED (*de novo*). Taxonomy-guided and assembly-guided queries are provided as Tables S3–S4 and S6–S7. (B) Comparison of spacers detected in the real AMD metagenome using Crass (*de novo*), MetaCRAST (using taxonomy-guided queries), MetaCRAST (using assembly-guided queries), and MinCED (*de novo*). Comparison was performed using Venny 2.1 (http://bioinfogp.cnb.csic.es/tools/venny/). (C) Comparison of spacers detected in the real EBPR metagenome using the same methods as in (B) Comparison was performed using Venny 2.1.

detection (Fig. 3). For AMD metagenomes simulated with the 454 model, MinCED detected significantly more false positive spacers than Crass or MetaCRAST for average read lengths of 200 bp or more (Fig. 6; $p < 0.05$ using unpaired $t$-tests). Crass and MetaCRAST, on the other hand, did not have statistically significant differences in detected false positive spacers over the entire range of average read lengths ($p > 0.05$ using unpaired $t$-tests). For the AMD Illumina metagenomes, on the other hand, MetaCRAST generated the largest number of false positive spacers for average read lengths greater than 200 bp (Crass for average read lengths of 150 bp and lower), but not by a statistically significant margin

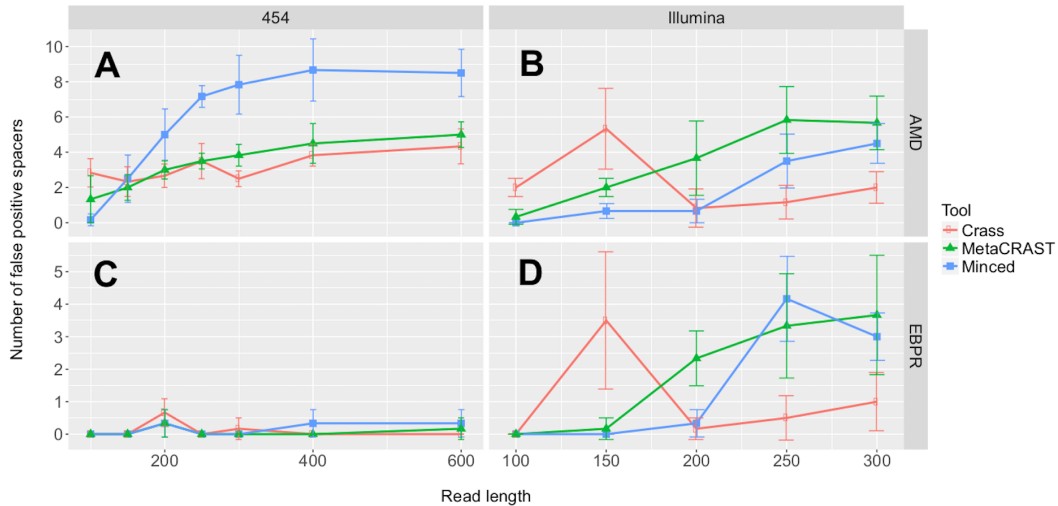

**Figure 6  Evaluation of MetaCRAST, Crass, and MinCED false positive detection on simulated AMD (A and B) and EBPR (C and D) metagenomes.** The procedure for generating the simulated metagenomes is described in Materials and Methods. The number of detected spacers matching expected ones was subtracted from the total number of spacers detected to determine the number of false positive spacers for a particular method and condition. All data points represent the averages of three individual simulations and are presented with error bars representing two times the standard error above and two below the average.

compared with MinCED ($p > 0.05$ using unpaired $t$-tests). For the EBPR metagenomes simulated with the 454 model, there were remarkably few false positive spacers detected with all methods over the full range of average read lengths. For the EBPR Illumina metagenomes, MinCED generated the largest number of false positive spacers for average read lengths greater than 200 bp (Crass for average read lengths of 150 bp and lower), with MetaCRAST overlapping its pattern closely (Fig. 6). Because of this overlap, differences between MinCED and MetaCRAST false positive spacers were not statistically significant ($p > 0.05$ using unpaired $t$-tests), (EBPR Illumina metagenomes, Fig. 6). MetaCRAST did detect more false positives than MinCED for the 200 bp read length ($p < 0.05$ using unpaired $t$-tests, EBPR Illumina metagenomes, Fig. 6). We note that these false positive spacers are only detected spacers that did not align to expected ones. The false positives do not necessarily include improperly truncated or extended spacers, which we suspect MinCED creates, leading to its artificially high spacer counts (Fig. 3). We repeated this false positive spacer analysis using a weaker $E$-value threshold of 1e−1 (Fig. S1). Using this weaker threshold decreased the number of false positive spacers identified in all conditions (Fig. S1).

We also evaluated relative speed of the detection methods using the Linux function *time*. We evaluated seven different combinations of algorithms, implementations, and parameters. We evaluated both Crass and MinCED with default parameters. For MetaCRAST, we evaluated five different conditions differing in parallelization and metagenome loading method—BioPerl for loading and 16 threads, BioPerl and a single thread, readfq with mce_open for loading and 16 threads, readfq with mce_open and a

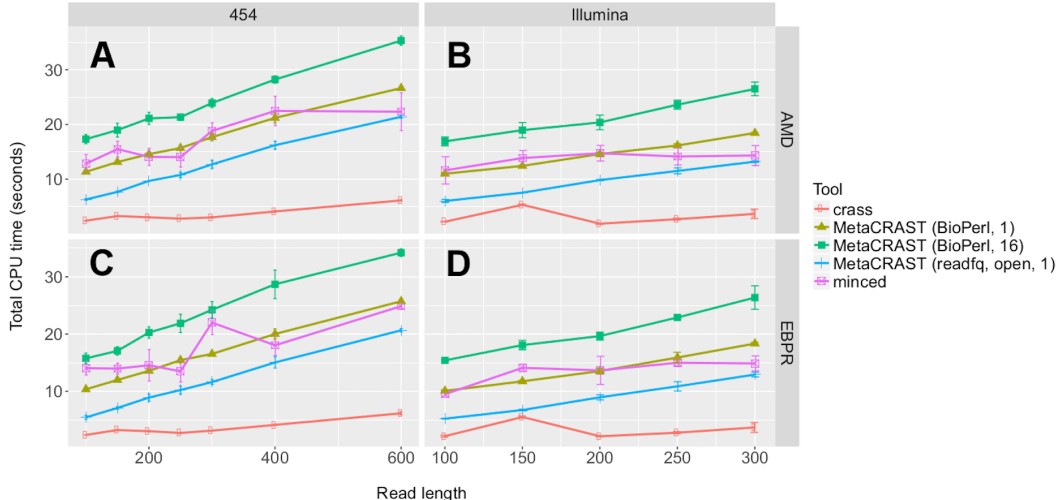

**Figure 7  Evaluation of MetaCRAST, Crass, and MinCED run times on simulated AMD (A and B) and EBPR (C and D) metagenomes.** We evaluated seven different combinations of algorithms, implementations, and parameters. We evaluated both Crass and MinCED with default parameters. For MetaCRAST, we evaluated five different conditions differing in parallelization and metagenome loading method—BioPerl loading and 16 threads, BioPerl and a single thread, readfq with mce_open for loading and 16 threads, readfq with mce_open and a single thread, and readfq with the standard open routine and a single thread. The procedure for generating the simulated metagenomes is described in Materials and Methods. Run time was calculated as the sum of the user and system time (together the total CPU time). All data points represent the averages of three individual simulations and are presented with error bars representing two times the standard error above and two below the average.

single thread, and readfq with the standard open routine and a single thread (Fig. 7). We used CPU time (user and system time) rather than wall clock time (real time) as a measure of speed performance.

We noticed steady increases in run time with increasing read length for all detection methods, metagenomes, and sequencing technologies (Fig. 7). MetaCRAST showed a linear CPU time dependence on read length in all cases ($R^2 > 0.98$ in all cases; $p$-values calculated from Pearson correlation were less than 1e−5 in all cases), while linear correlations for MinCED and crass were much weaker ($R^2 < 0.88$ in all cases; $p$-values calculated from the Pearson correlations were more than 0.05 for Illumina datasets but between 9e−4 and 8e−3 for 454 datasets). Among MetaCRAST implementations, the readfq/open version used the least CPU time by statistically significant margins for all conditions (Fig. 7; $p < 0.05$ in all cases using unpaired $t$-tests). MetaCRAST was slower than Crass for all read lengths by statistically significant margins (Fig. 7; $p < 0.05$ in all cases using unpaired $t$-tests). On the other hand, it was faster than MinCED for 454 read lengths between 100 and 400 bp and Illumina read lengths between 100 and 250 bp (Fig. 7; $p < 0.05$ using unpaired $t$-tests).

## Taxonomic affiliations of CRISPR direct repeats annotated in CRISPRdb

To analyze how direct repeats affiliated to taxa, we examined all direct repeats annotated in microbial genomes using the CRISPRdb database. We used a Perl script to assign taxonomy information based on GenBank accession using the module Bio::DB::GenBank. The results

of this analysis for species (binomial name) and genus-level designations are presented in Table 2. The average number of unique taxon designations per DR was greater at the species level than the genus level (1.308 compared to 1.063). Variation was also greater for species-level designations compared to genus-level (standard deviation of 1.567 compared 0.521). Both species- and genus-level analyses identified DRs that were affiliated to many taxa (a maximum of 20 genuses and 46 species). We acknowledge that our analysis does not examine the number of unique DRs per taxon. It also only considers independent, unique DRs, ignoring the possibility that many unique DRs may have closely related sequences.

## DISCUSSION

In this work, we present and evaluate a novel reference-guided method for CRISPR detection in unassembled metagenomic reads. This method searches metagenomic reads for user-specified direct repeats which could be provided through taxonomy- or assembly-guided searches (Figs. 1 and 2). We analyzed currently known DRs with respect to their taxonomic designations to determine the robustness of taxonomy-guided searches (Table 2). We found that most DRs in fact do affiliate to a single species or genus, but there are exceptions that may have arisen through horizontal gene transfer (Table 2). This analysis does not consider small polymorphisms between closely related DRs. Depending on the circumstance, it may be important to consider whether one DR could be present in multiple taxa found in a sample.

Our studies of simulated metagenomes show distinct advantages for Crass and MetaCRAST depending on average read length (Fig. 3). While the modified assembly procedure and exhaustive searches Crass provides make it well suited for short read 454 and Illumina metagenomes, MetaCRAST outperforms Crass for long read Illumina metagenomes (Fig. 3). We speculate that heuristics to avoid misassembly of CRISPR arrays or improper repeat detection may hinder Crass in these long-read Illumina metagenomes. We also noted that all three algorithms detected far more spacers in 454 compared to Illumina metagenomes (Fig. 3). We have two possible explanations for this phenomenon. First, our algorithms may have handled homopolymer error better than the substitution error simulated in the Illumina metagenomes. Second, our Illumina model may have introduced higher error rates than the 454 error model, making it more difficult to find multiple similar DRs in the reads. The very high numbers of MinCED-detected spacers are deceptive because this algorithm has the potential for substantial errors in determining repeat and spacer lengths (Figs. 3 and 4). Inconsistencies in defining repeat length leads to false splitting of identical spacers into different groups.

Studies on real metagenomes suggest substantial advantages for Crass and MinCED in terms of numbers of detected spacers (Fig. 5). While in most cases MetaCRAST detected fewer spacers than Crass or MinCED, it did identify spacers unique to those from the two other methods. This suggests that it can complement these methods, finding spacers missed due to the heuristics that Crass and MinCED use to avoid false positives (Fig. 5). We had expected that MetaCRAST would underperform compared to Crass and MinCED in these real metagenomes, because the taxonomy-guided queries we used did not fully

account for all the taxa found with taxonomic profiling. We only used one or two genomes to simulate the AMD and EBPR metagenomes, making the simulated metagenomes much simpler in taxonomic diversity. This simplification was what made MetaCRAST detection performance comparable to that of Crass and MinCED for the simulations.

Accuracy was roughly similar amongst the three tools (Fig. 6). Relaxing the error threshold reduced false positive spacers detected by all tools, suggesting sequencing error rather than algorithm issues could account for some of these false positive spacers (Fig. S1). MetaCRAST follows the same pattern of increasing run time with average read length as the other two tools, and it is comparable in run time to MinCED (Fig. 7). MetaCRAST run time increases linearly with average read length (Fig. 7). We acknowledge that implementation of the algorithm in a compiled language or increasing the number of threads used to parallelize the search could further improve MetaCRAST speed. Nonetheless, while MetaCRAST is not as fast as the compiled program Crass under the conditions tested, it does identify spacers distinct from these methods in real metagenomes and outperforms it in overall spacer detection for simulated Illumina metagenomes.

Recent studies of computational methods for determining phage-host interactions suggest CRISPR spacer alignment is a highly accurate signature of phage-host interaction but that most identified CRISPR spacers do not align to known phage genomes (*Edwards et al., 2015*). This suggests that it is critical to improve metagenomic CRISPR spacer detection to increase the chances of matching spacers to viral genomes. More broadly, increasing spacer matching would provide a fuller appreciation of a microbial ecosystem's phage-host interaction space. We have recently used MetaCRAST to improve our determination of virus-host interactions in solar salterns (*Moller & Liang, 2017*), complementing Crass with our spacer detection method. MetaCRAST complements *de novo* methods like Crass because it avoids the heuristics they use to reduce false positive spacers. Using a targeted direct repeat query, our tool can avoid the false negative bias of these approaches. We anticipate that MetaCRAST will be of great interest to microbial ecologists interested in phage-host interactions because it complements existing *de novo* methods to improve metagenomic CRISPR detection.

## ACKNOWLEDGEMENTS

Thanks to Michael Crowder and Gary Lorigan (Miami University) for feedback on the project and manuscript.

### Funding

The project was funded by the Committee on Faculty Research (CFR) program, the Office for the Advancement of Research & Scholarship (OARS), and by an Academic Challenge grant from the Department of Biology (Miami University). There was no additional external funding received for this study. The funders had no role in study design, data collection and analysis, decision to publish, or preparation of the manuscript.

### Grant Disclosures
The following grant information was disclosed by the authors:
Committee on Faculty Research (CFR) program.
Advancement of Research & Scholarship (OARS).
Department of Biology (Miami University).

### Competing Interests
Chun Liang is an Academic Editor for PeerJ.

### Author Contributions
- Abraham G. Moller conceived and designed the experiments, performed the experiments, analyzed the data, contributed reagents/materials/analysis tools, wrote the paper, prepared figures and/or tables, reviewed drafts of the paper.
- Chun Liang conceived and designed the experiments, contributed reagents/materials/-analysis tools, wrote the paper, prepared figures and/or tables, reviewed drafts of the paper.

### Data Availability
GitHub: https://github.com/molleraj/MetaCRAST.

### Supplemental Information
Supplemental information for this article can be found online at http://dx.doi.org/10.7717/peerj.3788#supplemental-information.

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
