# Peer review of "MetaCRAST: reference-guided extraction of CRISPR spacers from unassembled metagenomes"

_PeerJ, doi:10.7717/peerj.3788_

## Round 0.1 · original submission · Major Revisions

A number of questions were brought up by the reviewers - please address these concerns. Questions as to the creation and reliability of DR files for prediction of taxonomic source, data quality/error assumptions and overall assessment of performance should be particularly addressed.

Reviewer 1 ·

Basic reporting

This work proposed a Direct Repeat (DR) detection algorithm for metagenome CRISPRs. Please clarify the merits of the proposed algorithm since MetaCRAST runs slowest and seldom performs best for the dataset of read length 100 bps, which is the parameter of the major sequencing technology now.
1. A comparison is required to evaluation the proposed and other existing algorithms.
2. Recent literature demonstrated that the CRISPR DRs were under active recent mutations. Please clarify how such small-scale polymorphism was handled with specific parameters, and support the statements with appropriate references.
3. The spacer length is usually within a range, and please clarify the parameters used in this study.

Experimental design

No comment

Validity of the findings

No comment

Additional comments

No comment

Reviewer 2 ·

Basic reporting

The manuscript is written clearly but with many grammar errors, which need to be improved professionally in the revised version.

For example, in lines 280-283, “MetaCRAST complements de novo methods like Crass because it avoids the heuristics they use to reduce false positive spacers because it uses a targeted direct repeat query, it thus can avoid the false negative bias of these approaches.” This sentence “…because…because… ” was not written professionally and not easily understood. A number of sentences like this throughout the manuscript need to be rephrased.

Experimental design

To date, the maximum read length of Illumina data is 300 bps. Since there is no >300 bps Illumina data available, it’s impossible to evaluate whether these simulated data generated by Grinder are approximate or not to the real data. Thus, it’s not meaningful to compare the results of >300 bps between Illumina and 454 data. The authors need to revise Fig 3, Fig 6 and Fig 7, and make it clear in the main text and figure legends.

Validity of the findings

Firstly, in this study, the authors used simulated 454 data and Illumina data to validate the performance of the software, respectively. As shown in Fig3, under the same read length, only about half of the spacers were detected in Illumina data compared to those detected in 454 data by all three software. As stated in the manuscript, the difference between 454 and Illumina simulated data is the error model, i.e. the data quality. It has been well known that:
1. 454 data have artificial low quality homo-polymers
2. 454 data tend to have more sequencing errors

Thus it suggests that the data quality of short reads significantly affects the spacer detection. Unfortunately, the authors didn’t discuss it. In the revised version, the authors need to summarize the data quality of the simulated 454 and Illumina data with more details, and discuss why the error model affects the performance of all the three software. This would be very informative and benefit the scientific community to improve the software.

In the current version, MetaCRAST only takes fasta files as input, which ignores the data quality. Thus, it would potentially improve its performance if the software takes advantage of the base quality information of the data. The software needs to add one option to allow the fastq files (Illumina) or fasta + qual files (454) as input. In addition, to date, most of the whole-genome shotgun metagenomic data available in public databases are Illumina data, the usage of the sequence quality (fastq files) would not only improve the performance but also facilitate a lot of user.

Secondly, as shown in Fig 5A, in the performance comparisons based on the real data, MetaCRAST (using taxonomy-guided queries) was much worse than the de novo detection software Crass and MinCED, and MetaCRAST itself using assembly-guided queries. This is different from the comparisons based on the simulated data, where MetaCRAST performed close to Crass. Given that MetaCRAST detected some unique spacers, it’s hard to tell whether those are false positives or not. Thus, in the revised version, the authors need to further dig the data and identify the reasons why MetaCRAST missed the spacers that can be detected in de novo software and in the same software but on the assembled data. Again, it will improve the algorithm and the software.


Thirdly, in the figure legend of Figure 6, the authors stated in lines 479-481, “The number of detected spacers matching expected ones was subtracted from the total number of spacers detected to determine the number of false positive spacers for a particular method and condition.” However, it was not well explained in the manuscript how to identify the expected spacers in the simulated data, which directly affected the evaluation.

More importantly, as a reference-guided method, MetaCRAST identified spacers by searching reads containing reference DRs. Thus, different from the de novo detection methods, the false positive detection should be minimized in MetaCRAST. However, as shown in Figure 6, MetaCRAST detected the most numbers of false positive spacers in simulated Illumina data. This doesn’t make sense and needs to be discussed. This needs to be addressed whether it’s attributed to Wu-Manber multi-pattern search algorithm or other algorithm issues in MetaCRAST.

In lines 236-239, the authors stated, “We note that these false positive spacers are only detected spacers that did not align to expected spacers and thus do not necessarily include improperly truncated or extended spacers, which we suspect MinCED creates, leading to its artificially high spacer counts.” But if it was true for MinCED, how to explain false positive spacers detected by MetaCRAST?

In addition, the y-axis is confusing in Figure 6. If it represents number of false positive spacers, it should be integer. But why 454 data has 2.5 and 7.5 labeled in y-axis instead. This needs to clarify.

Fourthly, the statistical analysis should be added in the revised version of the study. For example, the authors stated, in lines 226-227, “Crass and MetaCRAST, on the other hand, did not have statistically significant differences in detected false positive spacers over the entire range of average read lengths.” What was the statistical test performed to claim “statistically significant differences” and what was the p-value?

In lines 228-231, the authors stated, “MetaCRAST generated the largest number of false positive spacers for average read lengths greater than 200 bp…, but not by a statistically significant margin compared with MinCED.” What was the statistical test performed and what was the p-value?

In lines 233-236, the authors stated, “MinCED generated the largest number of false positive spacers, …, with MetaCRAST overlapping its pattern closely.” Were they statistically different?

In lines 244-245, the authors stated, “MetaCRAST was the slowest for all read lengths, followed by MinCED and Crass, by statistically significant margins (Figure 7).” Again, What was the statistical test performed and what was the p-value?

Additional comments

In this study, Moller et al. developed a new reference-guided method to extract CRISPR spacers from metagenomic data, and compared its performance to other available ones. These comparisons are informative for the scientific community. However, there are several major issues needed be addressed and revised before publication.

·

Basic reporting

The paper describes a reference guided coputational method to detect CRISPR arrays in metagenomes. It has been available on github and as a preprint since 13/7/2016. MetaCRAST has been used in a recently accepted PeerJ paper by the same authors. As a preprint it has been downloaded 157 times. There are as yet no comments by ‘peers’.

The manuscript is well written and the approach is clearly useful, the literature is up to date to mid 2016. The software can be used in two ways firstly with a taxonomy-guided query or without this guide. The examples emphasize the taxonomy guided queries, but the more general assembly approach (using all known CRISPRdb spacers for example) would also be interesting.

Experimental design

The software can be installed easily locally following the instructions and runs as described and expected. The design is valid and a logical and useful extension of previous methods.

Validity of the findings

The analyses shown in the manuscript give a fair and convincing comparison to existing metagenomic CRISPR detection tools.

Additional comments

Major points
1. a. Describe how a fasta DR file with taxonomy information can be generated. There is a sentence in the methods stating this can be done (99-102) but without sufficient detail.
b. If the DR file is generated without knowledge of direction (as are those from CRISPRFinder/CRISPRdb described) then the reverse complement ‘-r’ should be used by default. In contrast those from CRISPRDetect have direction predicted (by CRISPRDirection).

2. If this DR file were to be used to predict the taxonomic source of the metagenomic (fasta) file being tested how reliable is this? The authors did this using MetaCRAST in a publication in PeerJ (Moller and Liang (2017), Determining virus-host interactions and glycerol metabolism profiles in geographically diverse solar salterns with metagenomics. PeerJ 5:e2844; DOI 10.7717/peerj.2844) but
how specific to each taxon is a DR? This point is difficult to address but the authors should discuss it, and/or analyse the DR input more fully.

a) If horizontal transfer of arrays occurs (as has been shown) different organisms could have the same DR. Some arrays are also found on mobile genetic elements (e.g. plasmids) so might easily be associated with multiple species.

For example, the DR file that can be obtained from CRISPRdb (as was done by the authors) contains a non-redundant set of DR, with many genomes listed in most fasta headers, some have over 50.
e.g. The first DR listing lists many genomes (NC.. or NZ..) for this DR.
>NC_000853_1|NC_000853_2|NC_000853_3|NC_000853_4|NC_000853_5|NC_000853_8|NC_009486_4|NC_009486_6|NC_009486_8|NC_009486_9|NC_010483_4|NC_010483_7|NC_010483_8|NC_013642_3|NC_013642_6|NC_013642_7|NC_013642_8|NC_023151_1|NC_023151_2|NC_023151_3|NC_023151_4|NC_023151_5|NC_023151_7|NZ_CP010967_1|NZ_CP010967_2|NZ_CP010967_3|NZ_CP010967_4|NZ_CP010967_8|NZ_CP011107_1|NZ_CP011107_2|NZ_CP011107_3|NZ_CP011107_4|NZ_CP011107_8|NZ_CP011108_1|NZ_CP011108_2|NZ_CP011108_3|NZ_CP011108_4|NZ_CP011108_8
GTTTCAATAATTCCTTAGAGGTATGGAAAC

b) Similarly, convergent evolution could give the same (or near identical) DR in different organisms. Some of the parameters -d, and –h specifically allow such DR mismatches as is appropriate. Have you tested appropriate values and can some be recommended? For example, at what range of edit distances is it MetaCRAST useful? Perhaps the analysis shown in Fig 4 could be extended to address this.

Notably, a few DR mismatches are permitted by other reference guided software (e.g. CRISPRDetect/CRISPRDiection) and this has proven useful in clustering DR.

3. The inputs are provided in the supplementary but not the command used. An additional file should be provided showing the parameters for each of these, if they vary from the defaults.

4. The manuscript dates from mid 2016 and in this fast-moving field should be brought up to date in the discussion. For example, CRISPRDetect (Biswas et al BMC Genomics 17:356 2016) uses a reference set of DRs to enhance predictions and direction of arrays in single genomes. Other papers have also addressed issues of false positives and reducing them using known DR (Zhang et al, BMC Bioinformatics 18:192 2017).

5. I do not fully see the value in clustering spacers, as is used in the test example, the value of using these should be explained in more detail.

---

## Round 0.2 · Minor Revisions

Thank you for addressing the reviewer concerns, addressing statistical issues, FastQ capability and the additional analysis.

It has been noted that there is an unacknowledged text overlap with your earlier article (https://peerj.com/articles/2844). Specifically, lines 44-50 of the current submission are direct from this earlier work. This overlap needs to be addressed before we can accept the manuscript.

One final note is that you may also want to rephrase the sentence on line 290 beginning with "We note that these false positive spacers..." as it is a bit confusing/awkward in reading.

·

Basic reporting

The authors have made substantial improvement to the manuscript, specifically addressing all my points, and carefully considering and addressing those of the other reviewers. I have no further concerns.

Experimental design

No comment

Validity of the findings

No comment

Additional comments

The authors have made substantial improvement to the manuscript, specifically addressing all my points, and carefully considering and addressing those of the other reviewers. I have no further concerns.

---

## Round 0.3 · accepted · Accept

Thank you for addressing the indicated concerns.